# Assessing the 10-Item Food Security Survey Model (FSSM): Insights from College Students in Three US Universities

**DOI:** 10.3390/nu17061050

**Published:** 2025-03-17

**Authors:** Rita Fiagbor, Onikia Brown

**Affiliations:** Department of Nutritional Science, Auburn University, 260 Lem Morrison, Auburn, AL 36849, USA; rnf0010@auburn.edu

**Keywords:** food insecurity, college students, cognitive interview, misinterpretation, survey

## Abstract

**Background/Objective**: Food insecurity remains a significant public health concern that negatively impacts college students’ academic performance and health. One in three college students experiences inconsistent access to food, known as food insecurity, which has attracted significant research interest. This study examined the effectiveness of the 10-item United States Department of Agriculture Food Security Scale Module (USDA-FSSM) in accurately and effectively measuring food security among college students. **Methods**: A mixed-methods approach was utilized to assess qualitative individual cognitive interviews and survey quantitative data. An online survey was used to collect demographic data and food security status from 462 college students recruited from three public universities in the United States. Qualitative interviews with a subset of participants (n = 26) were conducted to gain further insight into college students’ perceptions and interpretations of the 10-item USDA food security survey. **Results**: Fourteen (14%) participants were food-insecure, and 12% were at risk of food insecurity. Qualitative data revealed that students misinterpreted some of the language used in the 10-item USDA-FSSM. Participants also indicated difficulty estimating food security experiences over the 12-month reference period in the 10-item USDA-FSSM. **Conclusions**: This study demonstrates that college students misinterpret food security terms in the 10-item USDA-FSSM, which affects the prevalence rate determined by the measure, emphasizing the need for a validated college student-specific food security survey to inform effective policy and interventions.

## 1. Introduction

Food insecurity occurs when an individual or household lacks consistent access to adequate and nutritious food and/or is unable to obtain food in a socially acceptable manner [1,2,3]. The United States Department of Agriculture (USDA) indicated a steady rise in food insecurity among American households from 2021 to 2023. In 2023, 13.5 percent of households were food-insecure—a notable increase from 12.8 percent in 2022 and 10.5 percent in 2021 [4]. Food insecurity disproportionately affects several groups, often due to systemic inequalities and economic challenges. In fact, racial and ethnic minorities, households with low-income, and single-parent families are more vulnerable to food insecurity because of financial challenges and limited access to resources [5,6].

Food insecurity remains a significant public health concern that negatively impacts college students’ academic performance and health [2,7,8]. College years, a transitional period for young adults, further increase the risk of food insecurity due to limited financial resources; rising costs of tuition, housing, and food; time constraints; and lack of or restricted access to cooking facilities [9,10]. Despite declining college enrollment in the US between 2021 and 2022, racial minority enrollment slightly increased (45.23% in 2022 and 44.35% 2021) [11]. During the 2023–2024 admissions cycle, notable trends emerged among first-year students across various demographics. There was an 11% increase in applications from under-represented minority groups and a 5% increase in first-generation applicants, reflecting a broader push for diversity and accessibility in higher education in the US [12]. However, there are limited support systems, which has heightened food insecurity risk among college students [13,14].

Research on food insecurity among college students indicates that up to 45 percent of students experience some level of food insecurity at some point during their college years [15,16,17,18]. Limited access to adequate and nutritious food among college students adversely affects academic performance, retention, graduation rate, mental health, and overall well-being [2,3]. Traditional food security measures, like the USDA Food Security Survey Module (FSSM), were created and validated to evaluate household food security experiences. However, these measures do not account for the unique experiences, challenges, or lifestyles of college students [2,3,19,20].

One of the primary challenges in measuring food insecurity among college students is the variability in their financial resources and food acquisition strategies. Unlike the general population, college students may rely on meal plans, student organization meetings, food pantries, and irregular income sources, which influences their responses to standard FSSM items and the interpretation of survey responses [3,19,20]. Additionally, the concept of “money” in the context of food security may differ for students who receive financial aid or support from family members [19]. To address these challenges, researchers have suggested a need for a college-specific food security survey to better reflect the experiences of this population and incorporate items that account for the unique food access mechanisms in college environments [2,19,21,22,23].

Cognitive interviews can guide researchers to gain insight into how college students perceive the questions of the USDA-FSSM. Cognitive interviews have been implemented in previous study to inform and improve surveys [21,22]. Specifically, Nikolaus et al. evaluated the 6- and 10-item USDA-FSSM questions within one four-year university and concluded that the tool may not be the best to assess food insecurity among college students. To the best of our knowledge, the study published by Nikolaus et al. is the only other study that has assessed the fit of the USDA 10-item FSSM survey [21]. Thus, the objective of this study was to determine the effectiveness of the 10-item USDA-FSSM to accurately and effectively measure food security among college students. It was hypothesized that college students do not understand and misinterpret the 10-item food security survey, resulting in inaccurate food security status. This hypothesis was tested by comparing the survey responses with students’ answers provided in cognitive interviews about food security.

## 2. Materials and Methods

A mixed-methods study design was employed to collect information for this study. Quantitative assessment was conducted using a self-administered Qualtrics LLC (Provo, UT, USA) online survey to gather socio-demographic data (e.g., age, student status, gender, and ethnicity) of students from participating institutions. The online survey included the 10-item USDA Household Food Security Survey module, used to assess food security levels. The researchers conducted qualitative assessments using cognitive interviews with 26 college students to evaluate their understanding and interpretation of the food security survey items and provide suggestions to modify them to capture college students’ experiences and challenges. The researchers developed an interview guide that was used to guide the sessions, and the food security items were shared on a screen with participants during the interview session.

### 2.1. Sampling, Participant Recruitment, and Ethical Considerations

The study recruited college students from three land-grant institutions (Auburn University, University of Rhode Island, and South Dakota State University) using a student email registry, which provided access to enrolled students. The recruitment method primarily followed a convenience sampling approach, as all students with registered university emails had the opportunity to participate. In addition to being public universities, these universities provide a broad representation of college students across different regions, academic disciplines, and demographic backgrounds. The fact that these schools are public institutions with students from diverse populations, including students from rural areas, students from low-income households, and first-generation college students, can make their student bodies more prone to food insecurity (Table 1). To be eligible to participate in the study, the following criteria were necessary: enrolled as a college student, 18–24 years of age, and fluent in English. The invitation email included an information letter describing the study and a link to a 10-item survey on the Qualtrics LLC (Provo, UT, USA) survey platform.

The Institutional Review Boards of Auburn University, University of Rhode Island, and South Dakota State University approved the study protocol. Students accessed the survey information letter attached to the invitation email and at the beginning of the survey served as a virtual consent to participate in the online survey. However, before each virtual cognitive interview, each student gave verbal consent to indicate their willingness to participate and for the session to be recorded.

### 2.2. Procedure and Data Collection

The survey included ten (10) sociodemographic questions and the 10-item USDA Food Security Survey Module (FSSM) (Table 2). The sociodemographic characteristics collected in the survey included information on age, race, sex at birth, gender, sexuality, student enrollment status, student status, employment, family income bracket, and first-generation student status. Participants were instructed to complete the online Qualtrics LLC (Provo, UT, USA) survey.

Food security status was determined by the number of affirmative responses counted. An affirmative response was defined as answering ‘often’, ‘sometimes’, or ‘yes’ to the yes/no questions. The total number of affirmative responses formed each participant’s overall score. A score of 0 indicated high food security, while scores of 1–2 represented marginal food security. Scores of 3–5 indicated low food security, and any score above 6 signified very low food security. Therefore, a score of 3 or higher indicated food insecurity [21]. Upon completion, participants were directed to a separate platform to schedule a follow-up cognitive interview with a chosen pseudonym for anonymity.

Participants were categorized as food-secure or food-insecure based on responses to the 10-item FSSM from the survey before their cognitive interview. Food security status was not disclosed to the participant but was used in recruitment, data analysis, and reporting. At the start of the interview, participants were reminded to change their Zoom name to the previously selected pseudonym, and verbal consent was obtained to record the interview. The session proceeded once consent was given to record.

During the semi-structured, virtual interview, the 10-item FSSM served as a guide for the discussion, and the questions were shared on the screen with participants. The interviewer read each FSSM question and asked the participants to respond using think-aloud (thought process for answering the questions), comprehension, retrieval, confidence judgment, and response probes [27]. The interviewer used open-ended questions to elicit more detailed responses, clarification, and suggestions to modify survey items to capture college students’ experiences. Upon completion of the cognitive interview, participants were offered a USD 25 electronic gift card. Recruitment continued until data saturation was reached for both food-secure and food-insecure categories. The interview sessions were recorded via Zoom recording, then uploaded to a confidential, password-protected folder on Auburn University Box. All participants consented to the study electronically and verbally, and the study protocol was approved by the institutional review boards of participating schools.

### 2.3. Data Analysis

Quantitative and demographic data were analyzed and summarized using R statistical software version 4.4.1, with the level of significance established at *p < 0.05*. Descriptive statistics were used to evaluate the survey responses. Frequencies of food security status and demographic information were calculated. A chi-square analysis was conducted to assess the statistical association between the observed and expected values of sociodemographic and food security variables. As part of the chi-square analysis, standard residuals were calculated to identify where significant differences between observations and expectations occurred. In addition, ANOVA (Analysis of Variance) was used to compare mean differences in food security across various demographic groups.

The interview recordings were transcribed verbatim using Zoom transcription software (Version: 6.3.6 (56144)) and cleaned in a Microsoft Word document. The transcripts were reviewed and verified by two trained researchers and screened for themes, and codes were analyzed using ATLAS.ti version 22.2 (Scientific Software Development GmbH, based in Berlin, Germany).

## 3. Results

### Descriptive Findings

A total of 554 eligible undergraduate and graduate students from three US universities (Auburn, South Dakota, and Rhode Island) responded and completed the preliminary survey questionnaire on the Qualtrics platform. Survey responses were downloaded from the Qualtrics platform in Microsoft Excel format. Prior to analyzing the data, 92 responses were removed due to incomplete (i.e., responses with less than half of the 10-item USDA questionnaire answered) or duplicate responses, leaving a total of 462 responses. The data were then de-identified. The survey consisted of two parts: demographics and the 10-item USDA-FSSM questionnaire.

Sociodemographic data including age, gender, race, enrollment status, first-generation student status, family income brackets, and employment status were collected. Participants were mainly undergraduates (93%), with a mean age of 20.35 years old. Most of the students identified as female (76%) and Caucasian (76%). Most students reported they received financial support from family and friends. The majority (n = 256) identified their socioeconomic status as upper middle class, while about 6% of respondents were either not sure (n = 16) or preferred not to answer the question (n = 9). Among the study participants, 22.5% (n = 104) were first-generation students, 89.2% (n = 412) were enrolled in school full-time, and 63.4% (n = 293) indicated they worked part-time. Table 3 provides a detailed description of the study participants.

The responses to the food security survey were quantified. Food security status was categorized into high food security, marginal food security, and low food security based on the definition provided by the US Department of Agriculture [28]. Among students surveyed, 13.64% were food insecure and 12.34% were at risk of food insecurity (Figure 1).

Descriptive analyses revealed several factors influencing food insecurity among college students. Thirty percent of respondents indicated they could not afford a balanced diet, meaning they could not consume diverse, nutrient-rich foods. Additionally, 26% expressed concern that their food would run out before they had enough money to buy more. Twenty-two percent reported that the food they purchased did not last and they had no money to get more, highlighting a critical aspect of food insufficiency. Furthermore, 19% of participants reported cutting or skipping meals for various reasons, including financial constraints, with the same percentage eating less than they should. Lastly, 8% of participants reported not eating for an entire day (Table 4).

The chi-square test of independence was used to compare food security status with each sociodemographic factor within the study sample. There were significant associations between food security status and race, financial status, gender, sexuality, first-generation student status, and student status (*p < 0.001*). However, food security status was not significantly associated with college student employment status (*p > 0.05*). Findings suggest that multiple demographic factors, especially financial status (*p-value: 8.86E-09*) and student status (*p-value: 3.316E-07*), play a crucial role in determining food security status.

Analysis of the cognitive interview transcripts revealed the following themes: (1) Food accessibility and affordability, (2) understanding and awareness of nutritious food, (3) impact on health and well-being, (4) responses to food insecurity and dietary practices, and (5) survey comprehension and feedback. For the theme of food accessibility and affordability, the identified codes were meal plan affordability, skipping meals due to cost, reducing meal sizes, difficulties in obtaining healthy food, and eating the same unhealthy foods repeatedly due to budget constraints. For the theme of understanding and awareness of nutritious food, the identified codes were definitions of nutritious foods, awareness of food types and their health benefits, and the importance of a balanced diet. Illustrative quotes representing each theme and code are presented in Table 5.

Survey responses were summarized (Table 6) and compared with the interpretations of food security items—hunger, balanced meals, money for food, and time reference—to determine if content validity was compromised. The cognitive interviews revealed several key insights into college students’ interpretation of the 10-item FSSM questions. First, the phrase “money for food” was interpreted in various ways, including meal plans or dining dollars, paychecks for those working, part of their allowance allotted for food, or their ability to get groceries, which also encompassed money for food items and transportation. Students were asked to recall their food security experiences over the “last 12 months”; many found this challenging due to frequent relocations. For example, one student mentioned, “A little difficult, because with the pandemic, years have merged, hard to think that far back”, while another said, “It was difficult to remember the last 12 months because I lived in so many different places”. Students misinterpreted other key FSSM items, such as “household”, “balanced meals”, “hunger”, and “weight loss”.

Regardless of their food security status, students indicated that balanced meals were not a primary factor in their meal choices. Instead, they emphasized the time required for cooking, the availability of cooking facilities (which were often lacking in dorms), their ability to cook, and what was available in their pantry. Many preferred quick-fix meals or meals they could eat while driving, and some had to stretch their meals due to time constraints and lack of transportation for groceries. Regarding “skipping meals”, participants noted that factors beyond financial constraints, such as time constraints and emotional states, influenced this behavior. The term “household” was also problematic, as many students lived with dorm mates or in off-campus housing. For items requiring household food experience, students referred to their dorm mates or roommates.

## 4. Discussion

The objective of this study was to determine the effectiveness of the 10-item USDA-FSSM to accurately and effectively measure food security among college students. Cross-sectional data from 462 survey responses and 26 cognitive interviews were used to assess students’ comprehension of the 10-item FSSM in estimating food insecurity. Data supported both proposed hypotheses. The rate of food insecurity among a convenience sample of college students attending three publicly funded universities (Auburn University, South Dakota State University, and University of Rhode Island) was 13.6%, with an additional 12.34% at risk of food insecurity (Figure 1). These rates are comparable to household food insecurity in the United States (13.5%) [4] but within the lower range found at other universities [2,10,17,29,30,31,32]. Gaines et al. [33] found that 14% of college students at a large public university in the southeastern United States were food-insecure based on a sample of 557 undergraduate students. A study conducted at a university in the northeastern United States in spring 2017 (n = 1037) reported that 15% of students were food-insecure [32]. Additionally, Payne-Sturges et al. [2] found that 15% of their study sample of 237 undergraduate students at a public mid-Atlantic university were food-insecure. Although the prevalence of food insecurity in the present study is lower than the 19–54% reported in previous research, it still raises concerns about inconsistent access to food among college students [10,17,28,30,31,32]. The large range of prevalence further supports the need for a validated food security survey specifically for college students to capture students’ lived experiences and challenges [2,19,21,23].

Food insecurity status was not significantly associated with employment, implying current employment might not be adequate to cover basic needs. However, the findings do not reflect results from other studies, which suggest employment among students can significantly impact their food security status [2,17,34]. Another study suggested that the number of hours students work per week is associated with their level of food security [35]. In the current study, 63.4% reported working part-time, implying students work fewer hours and do not earn enough compared to those reported to work full-time or more hours [35]. Consistent with previous findings, this study found that students from ethnic minority backgrounds, undergraduates, and first-generation students were more likely to experience food insecurity [21,36]. The present study found that food security status, race, financial status, gender, sexuality, first-generation student status, and student status are associated.

Analysis of the interviews conducted as part of this study indicated college students often misinterpret FSSM questions, struggling with terms such as “money for food”, “balanced meals”, “household”, and “weight loss” in relation to insufficient food. Study data reported that 30% of college students did not consume balanced meals. This results in unmet nutritional needs, increasing their risk of chronic diseases and nutrient deficiencies. Students (26%) also worried their food would run out, suggesting uncertainty about their ability to consistently access enough food. This forces difficult choices between food and other basic needs, like housing, utilities, and healthcare. As a result, individuals may buy cheaper, less nutritious items to stretch their budget, potentially leading to poor dietary habits. It can also cause significant stress and anxiety, as individuals worry about where their next meal will come from. This psychological strain can affect mental well-being, contributing to anxiety, depression, or a sense of hopelessness, particularly if the situation persists over time, and may result in coping mechanisms like skipping meals or reducing portion sizes. It can increase pressure on community and social support services, such as food banks, charitable organizations, and government assistance programs. Other college students had concerns that the food items they purchased would not last long, highlighting challenges related to budgeting for groceries, the cost of food, and the sustainability of financial resources impacting their immediate diet, overall stability, and long-term health, emphasizing the need for individual and systemic interventions [21,29,37].

Furthermore, students reported varied interpretations of the term money for food. This complicates the term’s meaning, as it generally implies financial access to food in the broader population. The diversity in financial sources for college students, such as support from parents and friends or financial aid, influenced how this demographic viewed money for food. Similar to findings reported in the literature, college students often struggled to accurately recall their food security experiences over the past 12 months [3,38]. The time frames of the “last 12 months” and “last 30 days” have been validated among the general population, but the applicability of those time frames to college students might affect food insecurity rates [19,39,40].

Students suggested a modification of the 10-item USDA-FSSM to reflect the experiences and challenges of college students. The proposed areas include a reference time that reflects their schedules and the key food security terms. The present study contributes to the increasing recognition of a need for a validated food security survey specifically for college students to capture students’ lived experiences and challenges [2,19,21,23]

Limitations of this study include the use of a self-reported survey, which may not account for the honesty and accuracy of participants’ responses. Also, the mixed-methods approach employed in the study is labor- and time-intensive. Therefore, the number of students included in the qualitative investigation was small. However, data saturation was achieved for the cognitive interviews concerning the study objective. Additionally, a psychometric study indicated that the addition of a two-item food adequacy screener in the assessment of food security yields more accurate results. Hence, including qualitative investigations in these questions may be valuable. This study did not collect information on the portion of student income/allowance spent on food or other necessities, which could provide insight into whether certain behaviors are associated with food insecurity. Despite these limitations, this study reinforces previous research highlighting the necessity of a validated survey that accurately evaluates college students’ food insecurity and informs national and institutional food aid resource policies.

## 5. Conclusions

This study found evidence that college students often misunderstood the 10-item USDA Food Security Survey Module (FSSM). Thus, we can conclude the 10-item USDA-FSSM does not accurately and effectively measure food security among college students in the United States. This finding highlights the need for more targeted prevalence studies and a validated food security survey tailored to college students. The study also underscores the significant issue of food insecurity among college students, with 13.6% experiencing food insecurity and an additional 12.34% at risk. Although these values are lower than some reported in previous studies, they still emphasize the need for food aid interventions developed for college students. The findings suggest that student employment status does not necessarily mitigate food insecurity, and students from ethnic minority backgrounds, undergraduates, and first-generation students are particularly vulnerable. Misinterpretations of the food security survey questions and varied financial sources further complicate the assessment of food insecurity among college students. Addressing these challenges requires individual and systemic efforts to ensure consistent access to nutritious food, supporting overall well-being and academic success. Therefore, a college student food security assessment that reflects students’ interpretations of money for food, balanced diet, and household status is necessary to inform policies and strategies to decrease food insecurity among college students.

## Figures and Tables

**Figure 1 nutrients-17-01050-f001:**
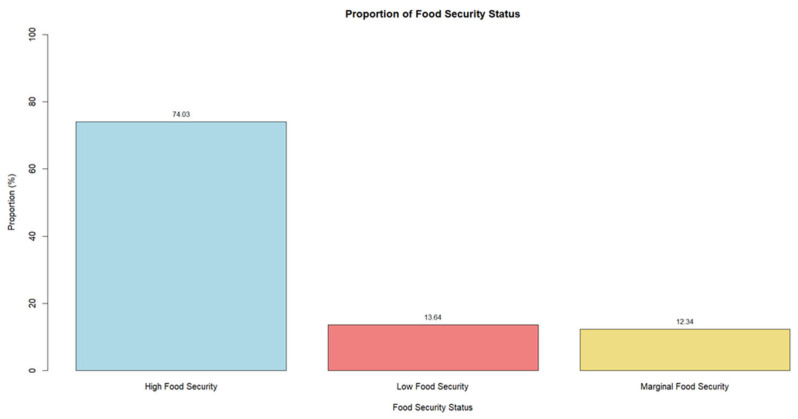
Food security status levels among college students based on the 10-item USDA Adult Food Security Survey Module used in the current study population. The chi-square test revealed a significant difference (*p-value < 2.2E-16*) between the observed distribution of food security status levels (high food security, marginal food security, and low food security).

**Table 1 nutrients-17-01050-t001:** Enrollment demographics of Auburn University, University of Rhode Island, and South Dakota State University [24,25,26].

	Auburn (n = 33,015) Serving 50 States and 107 Countries	Rhode Island (n = 17,352) Serving 47 States and 65 Countries	South Dakota (n = 12,065)-Serving 47 States and 77 Countries
Demographic Characteristics			
American Indian or Alaska Native	86 (0.3%)	35 (0.2%)	136 (1.1%)
Asian	985 (2.9%)	595 (3.4%)	158 (1.3%)
Black or African American	1628 (4.8%)	837 (4.8%)	157 (1.3%)
Hispanics of any race	1524 (4.5%)	1998 (11.5%)	358 (3%)
Native Hawaiian or other Pacific Islander	23 (0.06%)	10 (0.1%)	13 (0.1%)
Nonresident alien	1767 (5.2%)	495 (2.9%)	826 (6.8%)
Race and ethnicity unknown	122 (0.4%)	626 (3.6%)	398 (3.3%)
Two or more races	1048 (3.1%)	589 (3.4%)	262 (2.2%)
White	26,962 (79%)	12,167 (70.1%)	9758 (80.9%)

**Table 2 nutrients-17-01050-t002:** The 10-item USDA-FSSM items and coding of response options as food-secure or insecure based on a Current Population Survey (CPS). Similar scoring is often used to estimate food insecurity prevalence among college students.

Item	Affirmative (Insecure) Response(s)	Negative (Secure) Response(s)
10-item Food Security Survey Module:		
HH2. I worried whether my food would run out before I got money to buy more.	Often true, sometimes true	Never true, Do not know
HH3. The food that I bought just did not last, and I did not have enough money to get more.	Often true, sometimes true	Never true, Do not know
HH4. I could not afford to eat balanced meals.	Often true, sometimes true	Never true, Do not know
AD1. In the last 12 months, did you ever cut the size of your meals or skip meals because there was not enough money for food?	Yes	No, Do not know
AD1a. In the last 12 months, how often did this happen?	Almost every month, some months	N/A
AD2. In the last 12 months, did you ever eat less than you felt you should because there was not enough money for food?	Yes	No, Do not know
AD3. In the last 12 months, were you ever hungry but did not eat because there wasn’t enough money for food?	Yes	No, Do not know
AD4. In the last 12 months, did you lose weight because there was not enough money for food?	Yes	No, Do not know
AD5. In the last 12 months, did you ever not eat for a whole day because there was not enough money for food?	Yes	No, Do not know
AD5a. In the last 12 months, how often did this happen?	Almost every month, some months	N/A

**Table 3 nutrients-17-01050-t003:** Summary of participant sociodemographic characteristics.

	Total: n (%)
Age in years (mean ± SD)	462 (20.35 ± 1.55)
Gender at birth	
Male	105 (22.7%)
Female	353 (76.4%)
Non-binary	3 (0.7%)
Prefer not to say	1 (0.2%)
Gender identity/sexuality	
Heterosexual	399 (86.4%)
Bisexual	36 (7.8%)
Homosexual	11 (2.4%)
Other	4 (0.8%)
Prefer not to say	12 (2.6%)
Race	
Caucasian	349 (75.5%)
Black or African American	39 (8.4%)
Asian	22 (4.8%)
American Indian or Alaskan Native	7 (1.5%)
Native Hawaiian or Pacific Islander	4 (0.9%)
Other	33 (7.1%)
Choose not to answer	8 (1.7%)
Enrollment status	
Full-time student (12+ credit hours)	412 (89.2%)
Part-time student (under 12 credit hours)	45 (9.7%)
Choose not to answer	5 (1.1%)
School year	
Freshman	166 (35.9%)
Sophomore	90 (19.5%)
Junior	89 (19.3%)
Senior	83 (18.0%)
Graduate	31 (6.7%)
Choose not to answer	3 (0.6%)
First-generation college student	
Yes	104 (22.5%)
No	348 (75.3%)
Unsure	8 (1.7%)
Choose not to answer	2 (0.4%)
Family income bracket	
Lower class	21 (4.5%)
Lower middle class	144 (31.2%)
Upper class	16 (3.5%)
Upper middle class	256 (55.4%)
Do not Know	16 (3.5%)
Choose not to answer	9 (1.9%)
Employment status	
Full-time (36+ h per week)	26 (5.6%)
Part-time (1–36 h per week)	293 (63.4%)
I do not have a job.	137 (29.7%)
Choose not to answer	6 (1.3%)

**Table 4 nutrients-17-01050-t004:** Summary of FSSM survey item response.

Item	Affirmative Responses, n	Affirmative Responses, %
Worried about running out of food	120	26%
Purchased food did not last	101	22%
Cannot afford balanced meals	140	30%
Cut or skip meals	89	19%
Eat less than should	86	19%
Hungry, did not eat	77	17%
Lost weight	41	9%
Did not eat entire day	36	8%

**Table 5 nutrients-17-01050-t005:** Themes, codes, and illustrative quotes for interpretation of food security questions by college students.

Theme	Code	Illustrative Quote
Food accessibility and affordability	Meal plan affordability Skipping meals due to cost Reducing meal sizes Difficulties in obtaining healthy food Eating the same unhealthy foods repeatedly due to budget constraints	“I had to eat the same unhealthy thing for several days in a row because I didn’t have money to buy more.” “Even though we all have money saved up, it’s still hard because healthier foods tend to be more expensive.”
Understanding and awareness of nutritious food	Definitions of nutritious foods Awareness of food types and their health benefits Importance of a balanced diet	“I spent almost eight hours a day at school, and with the money I made from my job, I often couldn’t afford a nutritious, balanced meal. I also didn’t have enough time to pack a meal before leaving. So, when it came to getting food, I would just grab whatever was available like French fries. This only happened because I was at school. If I were at home, it would have been different. At home, I would have access to a sufficient amount of all food groups, not just focusing on one like carbs or protein.” “Balanced meals mean whatever the CDC recommends as the recommended portion of whatever type of food you eat each day—fruits, vegetables, grains, and protein. When I’m home, my mom eats insanely clean and always makes sure there’s enough healthy food in the house.”
Impact on health and well-being	Concern about the impact of food on health Feelings of wellbeing related to diet Long-term health concerns due to poor diet	“I have allergies, so the food I eat is often more expensive because it has to be a specific brand or type. This makes it really challenging for me to afford a balanced meal within my budget.” “I was hungry where I felt like I couldn’t function. I felt sick to my stomach. I felt nauseous, hungry to the point where I felt like I couldn’t go on anymore.”
Responses to food insecurity and dietary practices	Coping mechanisms for food scarcity Choices between unhealthy and healthy food when limited Behavioral responses to lack of food options	“I sometimes unintentionally cut the size of my meals, not necessarily because of money, but more like, ‘Oh, I only have half a box of pasta left, so I’ll make a smaller portion now, so I have enough for another meal later’”. “On a month-to-month basis, it changes. Within the past 12 months, there have been certain months where there was enough food, but not the kind I wanted to eat. However, there were other months where there simply wasn’t enough food at all”.
Survey comprehension and feedback	Clarity of survey questions Suggestions for improving questions Understanding and interpretation of survey prompts	“When you ask about my household, do you mean my household at school with roommates, or my household when I’m home with my family?” “For me, a balanced meal is something that has a solid amount of protein. Fat and carbs depend on what you’re looking for, but it should include some type of meat. That’s what I consider when thinking of a balanced meal”.

**Table 6 nutrients-17-01050-t006:** College students’ interpretation of the USDA-FSSM questionnaire—cognitive interview.

Phrase from Questionnaire Item	Interpretation	Sample Quote(s) from Interview
“Money for food” or any phrase associated with money	Money is interpreted differently due to heterogenous sources for college students, e.g., dining dollars, savings/allowance, employment status, and other support systems. Sources of finances were overlooked.	“I am a Finance major, money for food means monitoring my finances closely.” (Food-insecure female) “I work and my parents also support me.” (Food-secure female) “I always have money saved up, so I’m never worried about running out of money when it comes to groceries.” (Food-secure female)
“Balanced meals”	Balanced meals mean several things, e.g., money, preference, inability to cook, and time. There is a lack of understanding of the term, e.g., healthy foods are considered balanced.	“Balanced meals means some carbs in the meals as well as dealing with cravings.” (Food-secure female) “I think of balanced meals as the wheel chart (MyPlate), but I do not always stick to it not because I cannot afford but prefer quicker options.” (Food-insecure male). “Making balanced meals at school takes time, time to go shopping and make a meal.” (Food-secure female)
“Household”	- There is a lack of understanding of the term household. - Students exhibited difficulty in separating household at home, school, and other places lived during the period.	“A place of residence where I live for a long period of time such as my household at school at off-campus living situation.”. (Food-secure female) “My household is my home with my parents and siblings.” (Food-secure female) “My household is my dorm mate at school.” (Food-secure female)
“Cut the size of meals or skip meals”	- Varied reasons other than money were provided, e.g., time, stress from schoolwork, running low of dining dollars, and poor planning.	“I eat less than I should, especially getting to the end of the semester.” (Food-secure male) “I stretch out my meals so that they last longer, so I end up eating less.” (Food-insecure female) “It is just depending on my timing, sometimes I would skip meals, because I did not have enough time before leaving for school.” (Food-secure female)
“Hungry”	- Various interpretations were provided for hunger, including going days without food.	“For me hunger means not being able to eat more than once in a day.” (Food-secure male) “Hunger means not being able to think because I have not eaten.” (Food-secure female) “Real hungry that’s just like temporary; I would say, hunger would be like not eating for like a couple days.” (Food-secure female)
“Weight loss”	- Most students did not monitor their weight and only determined their weight changes based on the fit of their clothing. - Students who monitored their weight were either athletes or regulars at the gym. - Weight loss is treated as a touchy subject.	“Weight loss is a touchy subject; I don’t own a scale in my house because of obsession with the number.” (Food-secure female) “I work out a lot and I own a scale, so I am able to check.” (Food-secure female) “I have not lost weight because my clothes still fit.” (Food-secure male)
“Last 12 months”	- Students tended to overlook the last 12 months. - Students had difficulty remembering the last 12 months.	“It was difficult to remember the last 12 months because I lived in so many different places.” (Food-secure female) “A little difficult, because with the pandemic, years have merged together, hard to think that far back.” (Food-secure female)

## Data Availability

Data included in the article are referenced in the article.

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
