# Peer review of "Assessing the 10-Item Food Security Survey Model (FSSM): Insights from College Students in Three US Universities"

_nutrients, 2025, doi:10.3390/nu17061050_

Round 1
Reviewer 1 Report
Comments and Suggestions for Authors
Dear authors,
Please, correct the manuscript according to the following suggestions:
1- Authors affiliations: please, complete this space identifying the university, department, address, followed by email
L35-37 - Any updated (2024 and 2025)?
L48 - College students at which locations? Please, specify.
L56-67 - Are there other works recent published on the current efforts to assess food security among college students? If yes, include these items in the contextualization before the justification.
L141 - Include the software version, city, country.
L152 - Please, correct the typo for the word enrollment.
Figure 1 - Please, include statistical differences
L225 - Remove extra '%'
L226 - There is an extra '4'.
L226 - Which state? Please, indicate which table this information can be found.
L284-291 - The authors highlighted the limitation of study, which is relevant. Please, write how the results can contribute to support further research and the authorities positioning to decrease food insecurity across students.
Reviewer 2 Report
Comments and Suggestions for Authors
The paper “Assessing the 10-Item Food Security Survey Model (FSSM): Insights from College Students” takes up very important topics and contributes to the growth of literature for research (especially for nutritionists and economists) and producers of food for students.
Before the manuscript acceptation for publication in “Nutrients”, the following items should be revised:
The title
The analysis is based on a sample of selected college students who are not from all over the country. Therefore I suggest you provide the group characteristics in the subject, e.g. students from the state ...
Introducion
The authors did not propose research hypotheses to increase the work's scientific value.
Methods
What internet form was used for statistical Analysis?
I suggest you provide a sentence about the total number accepted for the study and how many completed the questionnaire (the study ended).
Statistical analysis
(p<0.05) - it should be Italic - p<0.05 - the same in the next ones
Results
Figure 1 and Table 3. There are no statistical significances entered (or their absence).
Was the question asked about what part of income is spent on food?
This is missing in the results and discussion of the results.
Conclusions
“Thus, the objective of this study was to deter mine the effectiveness of the 10-item USDA FSSM in accurately and effectively measuring food security among college students.” - the summary does not answer the question. -I suggest you add an appropriate sentence.
I suggest adding the summary conclusion - the strengths and weaknesses of the experience.
Reviewer 3 Report
Comments and Suggestions for Authors
This is a thoughtful study on food insecurity within a college student population at three land grant institutions. The investigation could be improved with attention to the following factors.
- Please provide more detail about the sample design and merits of including the land grant institutions that participated in the study. Some context justifying the inclusion of these specific institutions would be useful as well. My hope is that a more compelling and detailed argument about why the schools were used beyond convenience can be marshaled.
- Also, consider these schools in the broader universe of possibilities. Would land grant institutions be especially prone to food insecurity given the students they serve? What might that tell us about this study?
- I am unclear about how individual selection into the study was conducted. Randomization was used, but was this with all enrolled students and precisely when (especially in relation to the pandemic)? Did students self-select out of the possible study sample or did the researchers comb student records to avoid, say, minors completing the survey? What number of students on the list (every 10th student?) was selected from the list and was the randomization procedure the same across all institutions?
- I presume the study is adequately powered. Were power analyses conducted?
- The interviews are barely mentioned in the abstract and introduction but are actually an important study component. They may be among its most important contributions for validation and instrument respecification. Please underscore the value of the interviews earlier, especially if other studies have lacked such a component. A stronger case for study significance can be made because of the interviews but that’s an opportunity largely lost here. Make the best possible case for study significance.
- Should “Do not know” be coded as food security? If retaining this choice, please provide a stronger justification for this coding decision. Seems like “Do not know” might genuinely be a lack of recall, especially based on the interviews. So, make the case for coding it as you do.
- Could one more quote be added to each Illustrative Quote table section so each has two quotes? That would round them out and they could be called Illustrative Quotes.
- Given the results, especially from the interviews, what specific rewording choices would you recommend on the FSSM? Those could be put into the Discussion section, perhaps in a table with two columns: “Existing Measure” and “Recommended Measure.”
Well done with some room for improvement.
Round 2
Reviewer 2 Report
Comments and Suggestions for Authors
The authors have taken into account my suggestions. I have no more comments.
Comments on the Quality of English LanguageI have no comments on the language of the publication.